# LEARNING ONLINE DATA ASSOCIATION

## ABSTRACT

When an agent interacts with a complex environment, it receives a stream of percepts in which it may detect entities, such as objects or people. To build up a coherent, low-variance estimate of the underlying state, it is necessary to fuse information from multiple detections over time. To do this fusion, the agent must decide which detections to associate with one another. We address this data-association problem in the setting of an online filter, in which each observation is processed by aggregating into an existing object hypothesis. Classic methods with strong probabilistic foundations exist, but they are computationally expensive and require models that can be difficult to acquire. In this work, we use the deep-learning tools of sparse attention and representation learning to learn a machine that processes a stream of detections and outputs a set of hypotheses about objects in the world. We evaluate this approach on simple clustering problems, problems with dynamics, and a complex image-based domain. We find that it generalizes well from short to long observation sequences and from a few to many hypotheses, outperforming other learning approaches and classical non-learning methods.

## 1 INTRODUCTION

Consider a robot operating in a household, making observations of multiple objects as it moves around over the course of days or weeks. The objects may be moved by the inhabitants, even when the robot is not observing them, and we expect the robot to be able to find any of the objects when requested. We will call this type of problem *entity monitoring*. It occurs in many applications, but we are particularly motivated by the robotics applications where the observations are very high dimensional, such as images. Such systems need to perform online *data association*, determining which individual objects generated each observation, and *state estimation*, aggregating the observations of each individual object to obtain a representation that is lower variance and more complete than any individual observation. This problem can be addressed by an online recursive *filtering* algorithm that receives a stream of object detections as input and generates, after each input observation, a set of hypotheses corresponding to the actual objects observed by the agent.

When observations are closely spaced in time, the entity monitoring problem becomes one of *tracking* and it can be constrained by knowledge of the object dynamics. In many important domains, such as the household domain, temporally dense observations are not available, and so it is important to have systems that do not depend on continuous visual tracking.

A classical solution to the entity monitoring problem, developed for the tracking case but extensible to other dynamic settings, is a *data association filter* (DAF) (the tutorial of Bar-Shalom et al. (2009) provides a good introduction). A Bayes-optimal solution to this problem can be formulated, but it requires representing a number of possible hypotheses that grows exponentially with the number of observations. A much more practical, though much less robust, approach is a maximum likelihood DAF (ML-DAF), which commits, on each step, to a maximum likelihood data association: the algorithm maintains a set of object hypotheses, one for each object (generally starting with the empty set) and for each observation it decides to either: (a) associate the observation with an existing object hypothesis and perform a Bayesian update on that hypothesis with the new data, (b) start a new object hypothesis based on this observation, or (c) discard the observation as noise.

The engineering approach to constructing a ML-DAF requires many design choices, including the specification of a latent state space for object hypotheses, a generative model relating observations to objects, and thresholds or other decision rules for choosing, for a new observation, whether to

associate it with an existing hypothesis, use it to start a new hypothesis, or discard it. In any particular application, the engineer must tune all of these models and parameters to build a DAF that performs well. This is a time-consuming process that must be repeated for each new application.

A special case of entity monitoring is one in which the objects' state is static, and does not change over time. In this case, a classical solution is online (robust) clustering. Clustering algorithms perform data association (cluster assignment) an state estimation (computing a cluster center).

In this paper we explore training neural networks to perform as DAFs for dynamic entity monitoring and as online clustering methods for static entity monitoring. Although it is possible to train an unstructured RNN to solve these problems, we believe that building in some aspects of the structure of the DAF will allow faster learning with less data and allow the system to address problems with a longer horizon. We begin by briefly surveying the related literature, particularly focused on learning-based approaches. We then describe a neural-network architecture that uses self-attention as a mechanism for data association, and demonstrate its effectiveness in several illustrative problems. We find that it outperforms a raw RNN as well as domain-agnostic online clustering algorithms, and competitively with batch clustering strategies that can see all available data at once and with state-of-the-art DAFs for tracking with hand-built dynamics and observation models. Finally, we illustrate its application to problems with images as observations in which both data association and the use of an appropriate latent space are critical.

## 2 RELATED WORK

**Online clustering methods**     The typical setting for clustering problems is *batch*, where all the data is presented to the algorithm at once, and it computes either an assignment of data points to clusters or a set of cluster means, centers, or distributions. We are interested in the *online* setting, with observations arriving sequentially and a cumulative set of hypotheses output after each observation One of the most basic online clustering methods is *vector quantization*, articulated originally by Gray (1984) and understood as a stochastic gradient method by Kohonen (1995). It initializes cluster centers at random and assigns each new observation to the closest cluster center, and updates that center to be closer to the observation. Methods with stronger theoretical guaranteees, and those that handle unknown numbers of clusters have also been developed. Charikar et al. (2004) formulate the problem of online clustering, and present several algorithms with provable properties. Liberty et al. (2016) explore online clustering in terms of the facility allocation problem, using a probabilistic threshold to allocate new clusters in data. Choromanska and Monteleoni (2012) formulate online clustering as a mixture of separate expert clustering algorithms.

**Dynamic domains**     In the setting when the underlying entities have dynamics, such as airplanes observed via radar, a large number of DAFs have been developed. The most basic filter, for the case of a single entity and no data association problem, is the Kalman filter (Welch and Bishop, 2006). In the presence of data-association uncertainty the Kalman filter can be extended by considering assignments of observations to multiple existing hypotheses under the multiple hypothesis tracking (MHT) filter. A more practical approach that does not suffer from the combinatorial explosion of the MHT is the joint probabilistic data association (JPDA) filter, which keeps only one hypothesis but explicitly reasons about the most likely assignment of observations to hypotheses. Bar-Shalom et al. (2009) provides a detailed overview and comparison of these approaches, all of which require hand-tuned transition and observation models.

**Learning for clustering**     There is a great deal of work using deep-learning methods to find latent spaces for clustering complex objects, particularly images. Min et al. (2018) provide an excellent survey, including methods with auto-encoders, GANs, and VAEs. Relevant to our approach are *amortized inference* methods, including *set transformers* (Lee et al., 2018) and its specialization to *deep amortized clustering* (Lee et al., 2019), in which a neural network is trained to map directly from data to be clustered into cluster assignments or centers. A related method is *neural clustering processes* (Pakman et al., 2019), which includes an online version, and focuses on generating samples from a distribution on cluster assignments, including an unknown number of clusters.

**Visual data-association methods**     Data association has been explored in the context of visual object tracking (Luo et al., 2014; Xiang et al., 2015; Bewley et al., 2016; Brasó and Leal-Taixé, 2020; Ma et al., 2019; Sun et al., 2019; Frossard and Urtasun, 2018). In these problems, there is typically a fixed visual field populated with many smoothly moving objects. This is an important special case of the general data-association. It enables some specialized techniques that take advantage

of the fact that the observations of each object are typically smoothly varying in space-time, and incorporate additional visual appearance cues. In contrast, in our setting, there is no fixed spatial field for observations and they may be temporally widely spaced, as would be the case when a robot moves through the rooms of a house, encountering and re-encountering different objects as it does so. Our emphasis is on this long-term data-association and estimation, and our methods are not competitive with specialized techniques on fixed-visual-field tracking problems.

**Learning for data association**    There is relatively little work in the area of generalized data association, but Liu et al. (2019) provide a recent application of LSTMs (Hochreiter and Schmidhuber, 1997) to a rich version of the data association problem, in which batches of observations arrive simultaneously, with a constraint that each observation can be assigned to at most one object hypothesis. The sequential structure of the LSTM is used here not for recursive filtering, but to handle the variable numbers of observations and hypotheses. It is assumed that Euclidean distance is an appropriate metric and that the observation and state spaces are the same. Milan et al. (2017) combine a similar use of LSTM for data association with a recurrent network that learns to track multiple targets. It learns a dynamics model for the targets, including birth and death processes, but operates in simple state and observation spaces.

**Algorithmic priors for neural networks**    One final comparison is to other methods that integrate algorithmic structure with end-to-end neural network training. This approach has been applied to sequential decision making by Tamar et al. (2016), particle filters by Jonschkowski et al. (2018), and Kalman filters by Krishnan et al. (2015), as well as to a complex multi-module robot control system by Karkus et al. (2019). The results generally are much more robust than completely hand-built models and much more sample-efficient than completely unstructured deep-learning. We view our work as an instance of this general approach.

## 3    PROBLEM FORMULATION

The problem of learning to perform online data association requires careful formulation. When the DAF is executed online, it will receive a stream of input detections $z_1, \ldots z_T$ where $z_t \in \mathbb{R}^{d_z}$, and after each input $z_t$, it will output two vectors, $y_t = [y_{tk}]_{k \in (1..K)}$ and $c_t = [c_{tk}]_{k \in (1..K)}$, where $y_{tk} \in \mathbb{R}^{d_y}$, $c_{tk} \in (0, 1)$ and $\sum_k c_{tk} = 1$. The $y$ values in the output represent the predicted properties of the hypothesized objects and the $c$ values represent a measure of confidence in the hypotheses, in terms of the proportion of data that each one has accounted for. The maximum number of hypothesis "slots" is limited in advance to $K$. In some applications, the $z$ and $y$ values will be in the same space with the same representation, but this is not necessary.

We have training data representing $N$ different data-association problems, $\mathcal{D} = \{(z_t^{(i)}, m_t^{(i)})_{t \in (1..L_i)}\}_{i \in (1..N)}$, where each training example is an input/output sequence of length $L_i$, each element of which consists of a pair of input $z$ and $m = \{m_j\}_{j \in (1..J_t^{(i)})}$ which is a set of nominal object hypotheses representing the true *current state* of objects that have actually been observed so far in the sequence. It will always be true that $m_t^{(i)} \subseteq m_{t+1}^{(i)}$ and $J_t^{(i)} \leq K$.

Our objective is to train a recurrent computational model to perform DAF effectively in problems that are drawn from the same distribution as those in the training set. To do so, we formulate a model (described in section 4) with parameters $\theta$, which transduces the input sequence $z_1, \ldots, z_L$ into an output sequence $(y_1, c_1), \ldots, (y_L, c_L)$, and train it to minimize the following loss function:

$$\mathcal{L}(\theta; \mathcal{D}) = \sum_{i=1}^{N} \sum_{t=1}^{L_i} \mathcal{L}_{\mathrm{obj}}(y_t^{(i)}, m_t^{(i)}) + \mathcal{L}_{\mathrm{slot}}(y_t^{(i)}, c_t^{(i)}, m_t^{(i)}) + \mathcal{L}_{\mathrm{sparse}}(c_t^{(i)}) \ .$$

The $\mathcal{L}_{\mathrm{obj}}$ term is a *chamfer loss* (Barrow et al., 1977), which looks for the predicted $y$ that is closest to each actual $m_k$ and sums their distances, making sure the model has found a good, high-confidence representation for each true object:

$$\mathcal{L}_{\mathrm{obj}}(y, m) = \sum_{j} \min_{k} \frac{1}{c_k + \epsilon} \|y_k - m_j\| \ .$$

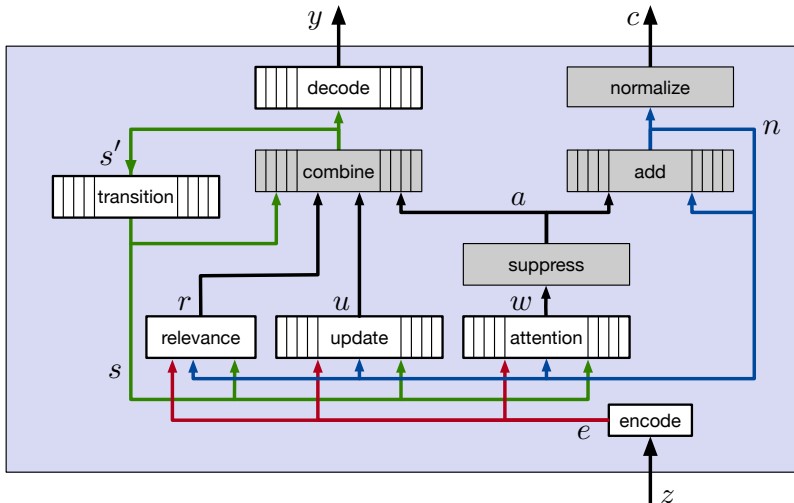

Figure 1: Architecture of the DAF-Net. Grey boxes represent fixed computations; white boxes represent neural networks with adjustable parameters; those with internal vertical bars represent a replication of the same computation on slot values in parallel. Red lines indicate information derived from an input observation, green lines indicate information derived some hypothesis slots values, and blue lines indicate information derived from counts on each hypothesis slot.

The $\mathcal{L}_{\text{slot}}$ term is similar, but makes sure that each object the model has found is a true object, where we multiply by $c_k$ to not penalize for predicted objects in which we have low confidence:

$$\mathcal{L}_{\text{slot}}(y, c, m) = \sum_k \min_j c_k \|y_k - m_j\| \ .$$

The sparsity loss discourages the model from using multiple outputs to represent the same true object:

$$\mathcal{L}_{\text{sparse}}(c) = -\log\|c\| \ .$$

## 4 DAF-NETS

Inspired by the the basic form of classic DAF algorithms and the ability of modern neural-network techniques to learn complex models, we have designed the DAF-Net architecture for learning DAFs and a customized procedure for training it from data, inspired by several design considerations. First, because object hypotheses must be available after each individual input and because observations will generally be too large and the problem too difficult to solve from scratch each time, the network will have the structure of a recursive filter, with new memory values computed on each observation and then fed back for the next. Second, because the loss function is *set based*, that is, it doesn't matter what order the object hypotheses are delivered in, our memory structure should also be permutation invariant, and so the memory processing is in the style of an attention mechanism. Finally, because in some applications the observations $z$ may be in a representation not well suited for hypotheses representation and aggregation, the memory operates on a latent representation that is related to observations and hypotheses via encoder and decoder modules.

Figure 1 shows the architecture of the DAF-Net model. There are six modules with adaptable weights and memory that is stored in two recurrent quantities, $s$ and $n$. The main memory is $s$, which consists of $K$ elements, each in $\mathbb{R}^{d_s}$; the length-$K$ vector $n$ of positive values encodes how many observations so far have been assigned to each slot. When an input $z$ arrives, it is immediately *encoded* into a vector $e$ in $\mathbb{R}^{d_s}$. The *update* network operates on the encoded input and the contents of each hypothesis slot, intuitively producing an update of the hypothesis in that slot under the assumption that the current $z$ is an observation of the object represented by that slot; so for all slots $k$,

$$u_k = \text{update}(s_k, n_k, e) \ .$$

The *attention* weights $w$ represent the degree to which the current input "matches" the current value:

$$w_k = \frac{\exp(\text{attend}(s_k, n_k, e))}{\sum_{j=0}^n \exp(\text{attend}(s_j, n_k, e))} \ .$$

To force the network to commit to a sparse assignment of observations to object hypotheses while retaining the ability to effectively train with gradient descent, the *suppress* module sets all but the top $M$ values in $w$ to 0 and renormalizes, to obtain the vector $a$ of $M$ values that sum to 1. The $a$ vectors are integrated to obtain $n$, which is normalized to obtain the final output confidence values $c$.

Additionally, a scalar *relevance* value, $r \in (0, 1)$, is computed from $s$ and $e$; this value is used to modulate the degree to which slot values are updated, and gives the machine the ability to ignore or downweight an input. It is computed as

$$r = \text{NN}_1(\underset{k=1}{\overset{K}{\text{avg}}} \, \text{NN}_2(e, s_k, n_k)) \ ,$$

where $\text{NN}_1$ is a fully connected network with the same input and output dimensions and $\text{NN}_2$ is a fully connected network with a sigmoid output unit. The attention output $a$ and relevance $r$ are now used to decide how to combine all possible slot-updates $u$ with the old slot values $s_t$ using the following fixed formula for each slot $k$:

$$s'_{tk} = (1 - ra_k)s_{tk} + ra_k u_k \ .$$

Because most of the $a_k$ values have been set to 0, this results in a sparse update which will ideally concentrate on a single slot to which this observation is being "assigned."

To compute the outputs, the $s'_t$ slot values are decoded into the representation that is required for the outputs, $y$:

$$y_k = \text{decode}(s'_{tk}) \ .$$

Finally, to handle the setting in which object state evolves over time, we can further add a dynamics model, which computes the state $s_{t+1}$ from the new slot values $s'_t$ using an additional neural network:

$$s_{t+1 \, k} = \text{NN}_3(s'_t)_k \ .$$

These values are fed back, recurrently, as inputs to the overall system.

Given a data set $\mathcal{D}$, we train the DAF-Net model end-to-end to minimize loss function $\mathcal{L}$, with a slight modification. We find that including the $\mathcal{L}_{\text{sparse}}$ term from the beginning of training results in poor learning, but adopting a training scheme in which the $\mathcal{L}_{\text{sparse}}$ is first omitted then reintroduced over training epochs, results in reliable training that is efficient in both time and data.

## 5 EMPIRICAL RESULTS

We evaluate DAF-Net on several *entity monitoring* tasks, including simple online clustering, monitoring objects with dynamics, and high-dimensional image pose prediction in which the observation space is not the same as the hypothesis space. Our experiments aim to substantiate the following claims:

- DAF-Net outperforms non-learning clustering methods, even those that operate in batch mode rather than online, because those methods cannot learn from experience to take advantage of information about the distribution of observations and true object properties (tables 1, 2 and 5).
- DAF-Net outperforms clustering methods that can learn from previous example problems when data is limited, because it provides useful structural bias for learning (table 1, 2 and 5).
- DAF-Net generalizes to differences between training and testing in (a) the numbers of actual objects, (b) the numbers of hypothesis slots and (c) the number of observations (tables 1 and 3).
- DAF-Net works when significant encoding and decoding are required (table 5).
- DAF-Net is able to learn dynamics models and observation functions for the setting when the entities are moving over time (table 4), nearly matching the performance of strong data association filters with known ground-truth models.

We compare with the following alternative methods: **Batch, non-learning**: K-means++ (Arthur and Vassilvitskii, 2007) and expectation maximization (EM) (Dempster et al., 1977) on a Gaussian mixture model (SciKit Learn implementation); **Online, non-learning**: vector quantization (Gray, 1984); **Batch, learning**: set transformer (Lee et al., 2018); **Online, learning**: LSTM (Hochreiter and Schmidhuber, 1997) and an online variant of the set transformer (Lee et al., 2018); **Dynamic, non-learning**: joint probabilistic data association filter (Bar-Shalom et al., 2009). All learned network

| Model | Online | Learned | Observations | | | |
|---|---|---|---|---|---|---|
| | | | 10 | 30 | 50 | 100 |
| DAF-Net | + | + | **0.235 (0.001)** | **0.162 (0.001)** | **0.146 (0.001)** | **0.128 (0.001)** |
| Set Transformer | + | + | 0.390 (0.002) | 0.388 (0.002) | 0.388 (0.002) | 0.389 (0.001) |
| LSTM | + | + | 0.288 (0.001) | 0.260 (0.001) | 0.269 (0.001) | 0.288 (0.001) |
| VQ | + | - | 0.246 (0.001) | 0.172 (0.001) | 0.147 (0.001) | 0.122 (0.001) |
| Set Transformer | - | + | 0.295 (0.003) | 0.261 (0.001) | 0.253 (0.001) | 0.247 (0.001) |
| K-means++ | - | - | **0.183 (0.002)** | **0.107 (0.001)** | **0.086 (0.001)** | **0.066 (0.001)** |
| GMM | - | - | 0.189 (0.002) | 0.118 (0.001) | 0.087 (0.001) | 0.067 (0.001) |

Table 1: Comparison of performance after training on one thousand *Normal* distributions for a thousand iterations. We use 3 components, and train models with 30 observations. We report standard error in parentheses. Each cluster observation and center is drawn between -1 and 1, with reported error as the L2 distance between predicted and ground truth mean.

| Model | Online | Learned | Normal | Elongated | Mixed | Angular | Noise |
|---|---|---|---|---|---|---|---|
| DAF-Net | + | + | **0.157 (0.001)** | **0.191 (0.001)** | **0.184 (0.001)** | **0.794 (0.001)** | **0.343 (0.001)** |
| Set Transformer | + | + | 0.407 (0.001) | 0.395 (0.001) | 0.384 (0.001) | 0.794 (0.003) | 0.424 (0.001) |
| LSTM | + | + | 0.256 (0.001) | 0.272 (0.001) | 0.274 (0.001) | 0.799 (0.002) | 0.408 (0.001) |
| VQ | + | - | 0.173 (0.002) | 0.195 (0.002) | 0.191 (0.002) | 0.992 (0.004) | 0.947 (0.002) |
| Set Transformer | - | + | 0.226 (0.001) | 0.248 (0.001) | 0.274 (0.001) | **0.816 (0.001)** | **0.406 (0.002)** |
| K-means++ | - | - | **0.103 (0.001)** | **0.139 (0.001)** | **0.135 (0.001)** | 0.822 (0.003) | 1.259 (0.002) |
| GMM | - | - | 0.113 (0.001) | 0.141 (0.001) | 0.136 (0.001) | 0.865 (0.003) | 1.207 (0.002) |

Table 2: Comparison of performance on clustering after 30 iteration when training on 1000 different distributions for a thousand iterations. We use a total of 3 components, and train models with 30 observations. We report standard error in parentheses. Each cluster observation and center is drawn between -1 and 1, except for angular which is drawn between $-\pi$ and $pi$ with reported error as the L2 distance between predicted and ground truth mean.

architectures are set to have about 50000 parameters. We provide additional details about architecture and training in the appendix. The set transformer is a standard architecture that has been evaluated on clustering problems in the past.

All models except DAF-Net are given the ground truth number of components $K$, while DAF-Net uses 10 hypothesis slots. Results are reported in terms of loss $\sum_j \min_k \|y_k - m_j\|$ (with the most confident $K$ hypotheses selected for DAF-Net).

**Gaussian domains** To check the basic operation of the model and understand the types of problems for which it performs well, we tested in simple clustering problems with the same input and output spaces, but different types of data distributions, each a mixture of three components. We train on 1000 problems drawn from each problem distribution distribution and test on 5000 from the same distribution. In every case, the means of the three components are drawn at random for each problem.

1. *Normal*: Each component is a 2D Gaussian with fixed identical variance across each individual dimension and across distributions. This is a basic "sanity check."
2. *Elongated*: Each component is a 2D Gaussian, where the variance along each dimension is drawn from a uniform distribution, but fixed across distributions.
3. *Mixed*: Each component is a 2D Gaussian, with fixed identical variance across each individual dimension, but with the variance of each distribution drawn from a uniform distribution.
4. *Angular*: Each component is a 2D Gaussian with identical variance across dimension and distribution, but points above $\pi$ are wrapped around to $-\pi$ and points below $-\pi$ wrapped to $\pi$
5. *Noise*: Each component has 2 dimensions parameterized by Gaussian distributions, but with the values of the remaining 30 dimensions drawn from a uniform centered at 0.

We compare our approach to each of the non-dynamic baselines for the five problem distributions in Table 1; a complete listing of results for all the distributions can be found in the Appendix. The results in this table show that on *Normal*, *Mixed*, and *Elongated* tasks, DAF-Net performs comparably to the offline clustering algorithms, even though it is running and being evaluated online. On the *Angular* and *Noise* tasks, DAF-Net is able to learn a useful metric for clustering and outperforms both offline

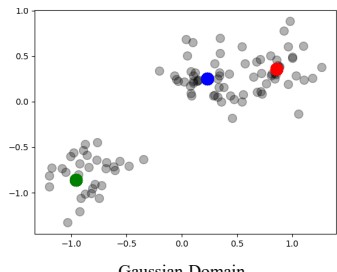 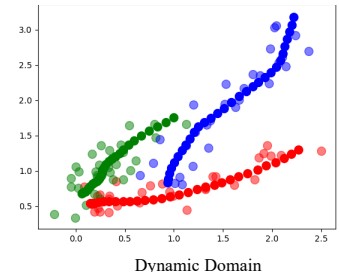

Gaussian Domain                    Dynamic Domain

Figure 2: Visualizations of Dynamic and Gaussian domains. Observations are transparent while while ground truth states are bolded

| Model | Slots | Ground Truth Clusters | | |
|---|---|---|---|---|
| | | 3 | 5 | 7 |
| DAF-Net | 10 | **0.162 (0.001)** | 0.214 (0.001) | 0.242 (0.001) |
| | 20 | 0.175 (0.001) | **0.195 (0.001)** | 0.213 (0.001) |
| | 30 | 0.188 (0.001) | 0.197 (0.001) | **0.205 (0.001)** |
| Set Transformer | - | 0.261 (0.001) | 0.279 (0.001) | 0.282 (0.001) |
| Vector Quantization | - | 0.171 (0.001) | 0.199 (0.001) | **0.205 (0.001)** |

Table 3: Quantitative evaluation of DAF-Net on distributions with different numbers of true components and hypothesis slots *at test time* with 30 observations. In all cases, DAF-Net is trained with 3-component problems, 10 slots, and 30 observations. We compare with an offline set transformer trained with different numbers of problem components as well as with vector quantization.

and online alternatives (with additional analysis in the appendix showing DAF-Net outperforms all other learning baselines with more training distributions in the *Angular* task).

In Table 1 we evaluate the quality of predictions after 10, 30, 50, and 100 observations in the *Normal* distribution. We find that DAF-Net generalizes well to increased numbers of observations, with predictions becoming more accurate as the observation sequence length increases, despite the fact that it is trained only on observation sequences of length 30. This is in contrast with other online learning baselines, set transformer and LSTM, which both see increases in error after 50 or 100 observations. This pattern holds across all the test problem distributions (see Appendix).

In Table 3, we investigate the generalization ability of DAF-Net to both increases in the number of hypothesis slots and the underlying number of mixture components from which observations are drawn. We compare to the offline set transformer and to VQ, both of which know the correct number of components at test time. Recall that, to evaluate DAF-Net even when it has a large number of extra slots, we use its $K$ most confident hypotheses. We find that DAF-Net generalizes well to increases in hypothesis slots, and exhibits improved performance with large number of underlying components, performing comparably to or better than the VQ algorithm. We note that none of the *learning* baselines can adapt to different numbers cluster components at test time, but find that DAF-Net outperforms the set transformer even when it is trained on the ground truth number of clusters in the test. We also ablated each component of our model and found that each of our proposed components enables both better performance and generalization. Detailed results of the ablations and a figure illustrating the clustering process are in the appendix.

**Dynamic Domains**    We next evaluate the ability of DAF-Net to perform data association in domains where objects are moving dynamically over time. This domain is typical of tracking problems considered by data association filters, and we compare with the de-facto standard method, Joint Probabilistic Data Association (JPDA), which uses hand-built ground-truth models. We consider a setup consisting of 3 different moving objects in 2D. Their velocity is perturbed at each step by an additive component drawn from a Gaussian distribution and observations of their positions (but no observations of velocities) are made with Gaussian error. To perform well in this task, a model must discover that it needs to estimate the latent velocity of each object, as well as learn the underlying dynamics and observation models.

| Model | Observations | | | |
|---|---|---|---|---|
| | 10 | 20 | 30 | 40 |
| DAF-Net | **0.322 (0.009)** | **0.187 (0.007)** | **0.168 (0.008)** | **0.195 (0.014)** |
| Online Set Transformer | 4.588 (0.006) | 4.499 (0.005) | 4.42 (0.006) | 4.43 (0.005) |
| LSTM | 0.348 (0.007) | 0.390 (0.011) | 0.506 (0.011) | 0.687 (0.018) |
| JPDA (ground truth) | **0.316 (0.022)** | **0.157 (0.016)** | **0.142 (0.008)** | **0.141 (0.002)** |

Table 4: Comparison of performance on position estimation of 3 dynamically moving objects. All learning models are trained with 1000 sequences of 30 observations. We report standard error in parentheses. JPDA uses the ground-truth observation and dynamics models.

| Type | Model | Learned | Observations | | | |
|---|---|---|---|---|---|---|
| | | | 10 | 30 | 50 | 100 |
| MNIST | DAF-Net | + | **7.143 (0.006)** | **5.593 (0.004)** | **5.504 (0.004)** | **5.580 (0.004)** |
| | LSTM | + | 9.980 (0.005) | 9.208 (0.004) | 9.166 (0.004) | 9.267 (0.004) |
| | K-means (Pixel) | - | 13.214 (0.005) | 12.010 (0.005) | 11.961 (0.004) | 11.719 (0.004) |
| | K-means (Learned) | + | 13.596 (0.006) | 12.505 (0.005) | 12.261 (0.003) | 12.021 (0.004) |
| Airplanes | DAF-Net | + | **4.558 (0.005)** | **4.337 (0.004)** | **4.331 (0.004)** | **4.325 (0.004)** |
| | LSTM | + | 5.106 (0.003) | 4.992 (0.005) | 4.983 (0.003) | 4.998 (0.004) |
| | K-means (Pixel) | - | 7.127 (0.006) | 6.890 (0.004) | 6.603 (0.004) | 6.517 (0.004) |
| | K-means (Learned) | + | 7.246 (0.006) | 6.943 (0.005) | 6.878 (0.005) | 6.815 (0.004) |

Table 5: Comparison of performance of online clustering on MNIST and on rendered Airplane dataset. For DAF-Net, LSTM and K-means (Learned) we use a convolutional encoder/decoder trained on the data; for K-means (Pixel) there is no encoding. We use a total of 3 components and train models with 30 observations. Models are trained on 20000 problems on both datasets.

We compare our approach to the Set Transformer and LSTM methods, as well as to JPDA with ground-truth models. The basic clustering methods have no ability to handle dynamic systems so we omit them from the comparison. The learning methods (DAF-Net, Set Transformer, and LSTM) are all trained on observation sequences of length 30. We test performance of all four methods on sequences of multiple lengths. Quantitative performance, measured in terms of prediction error on true object locations, is reported in Table 4. We can see that the online Set Transformer cannot learn a reasonable model at all. The LSTM performs reasonably well for short (length 10) sequences but quickly degrades relative to DAF-Net and JPDA as sequence length increases. We note that DAF-Net performs comparably to but just slightly worse than JPDA. This is very strong performance because DAF-Net is generic and can be adapted to new domains given training data without the need to hand-design the models used by JPDA.

**Image-based domains** We now evaluate the ability of DAF-Net to perform data association in domains with substantially more complex observation spaces, where the outputs are not simple averages of the inputs. This requires the network to synthesize a latent representation for slots in which the simple additive update performs effectively.

We investigate this with two image-based domains. In each domain, we have a set of similar objects (digits or airplanes). A *problem* is constructed by selecting $K$ objects from the domain, and the desired $y$ values are images of those objects in a canonical viewpoint. The input observation sequence is generated by randomly selecting one of those $K$ objects, and then generating an image of it from a random viewpoint as the observation $z$.

Our two domains are: (1) **MNIST**: Each object is a random digit image in MNIST, with observations corresponding to that same image rotated, and the desired outputs being the un-rotated images; (2) **Airplane**: Each object is a random object from the Airplane class in ShapeNet (Chang et al., 2015), with observations corresponding to airplane renderings (using Blender) at different viewpoints and the desired outputs the objects rendered in a canonical viewpoint.

For MNIST, we use the 50000 digit images in the training set to construct the training problems, and the 10000 images in the non-overlaping test set to construct the test problems. For the Airplane dataset, we use 1895 airplanes objects to construct the training problems, and 211 different airplanes objects to construct the test problems. Each object is rendered with 300 viewpoints. Of our baseline methods, only batch K-means (in pixel space) can be directly applied to this problem with even

Figure 3: Results on two image-based association tasks (left: MNIST, right: airplanes). At the top of each is an example training problem, illustrated by the true objects and an observation sequence. Each of the next rows shows an example test problem, with the ground truth objects and decoded slot values. The three highest-confidence hypotheses for each problem are highlighted in red, and correspond nicely to the ground-truth objects.

reasonable results. We also include versions of LSTM and of batch K-means that operate on a latent representation that is learned first using an auto-encoder. In Table 5, we find that our approach significantly outperforms other comparable baselines in both accuracy and generalization. We visualize qualitative predictions from our model in Figure 3.

# 6 DISCUSSION

This work has demonstrated that using algorithmic bias inspired by a classical solution to the problem of filtering to estimate the state of multiple objects simultaneously, coupled with modern machine-learning techniques, we can arrive at solutions that learn to perform and generalize well from a comparatively small amount of training data.

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

## A.1 APPENDIX

## A.2 DISCOVERY OF OBJECTS

In contrast to other algorithms, DAF-Net learns to predict both a set of object properties $y_k$ of objects and a set of confidences $c_k$ for each object. This corresponds to the task of both predicting the number of objects in set of observations, as well as associated object properties. We evaluate the ability to regress object number in DAF-Net in scenarios where the number of objects is different than that of training. We evaluate on the Normal distribution with a variable number of component distributions, and measure inferred component through a threshold confidence. DAF-Net is trained on a dataset with 3 underlying components.

We find in Figure A1 that DAF-Net is able to infer the presence of more component distributions (as they vary from 3 to 6), with improved performance when cluster centers are sharply separated (right figure of Figure A1).

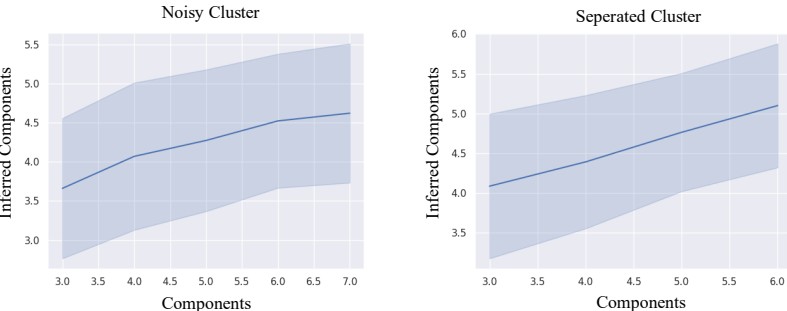

Figure A1: Plots of inferred number of components using a confidence threshold in DAF-Net compared to the ground truth number of clusters (DAF-Net is trained on only 3 clusters). We consider two scenarios, a noisy scenario where cluster centers are randomly drawn from -1 to 1 (left) and a scenario where all added cluster components are well seperated from each other (right). DAF-Net is able to infer more clusters in both scenarios, with better performance when cluster centers are more distinct from each other.

## A.3 QUALITATIVE VISUALIZATIONS

We provide an illustration of our results on the *Normal* clustering task in Figure A2. We plot the decoded values of hypothesis slots in red, with size scaled according to confidence, and ground-truth cluster locations in black. DAF-Net is able to selectively refine slot clusters to be close to ground truth cluster locations even with much longer observation sequences than it was trained on.

We find that each component learned by DAF-Net is interpretable. We visualize attention weights of each hypothesis slot in Figure A3 and find that each hypothesis slot learns to attend to a local region next to the value it decodes to. We further visualize a plot of relevance weights in Figure A4 across increasing number of observations over different levels of noise in each distribution. We find that as

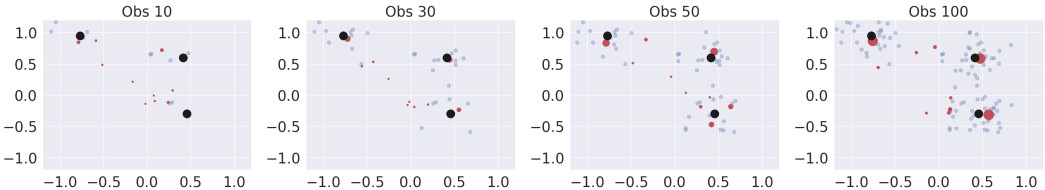

Figure A2: Illustration of the clustering process. Decoded value of hypothesis (with size corresponding to confidence) shown in red, with ground truth clusters in black. Observations are shown in blue.

we see more observations, the relevance weight of new observations decreases over time, indicating that DAF-Net learns to pay the most attention towards the first set of observations it sees. In addition, we find that in distributions with higher variance, the relevance weight decreases more slowly, as later observations are now more informative in determining cluster centers.

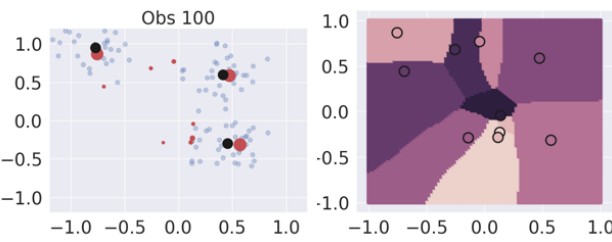
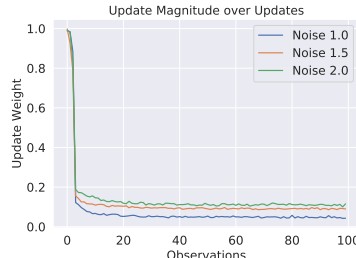

Figure A3: Plot of slots (left), and what slot each input assigns the highest attention towards (right) (each slot is colored differently, with assigned inputs colored in the same way). Note alignment of regions on the right with point density on the left.

Figure A4: Plots of the magnitude of relevance weights with increased observation number on different distributions with higher standard deviation (noise).

## A.4 QUANTITATIVE RESULTS

We report full performance of each different model across different distributions in Table 6. We find that DAF-Net is able to obtain better performance with increased number of observations across different distributions. In addition DAF-Net out-performs neural network baselines when evaluated on 30 observations across distributions except for rotation. For rotation we find that when training with 10,000 different distribution, DAF-Net exhibits better performance of 0.555 compared to Set Transformer Online performance of 0.647 and LSTM performance of 0.727.

## A.5 SPARSITY LOSS

In this section, we show that $\mathcal{L}_{\text{sparse}}(\mathbf{c})$ encourage confidences $\mathbf{c}$ to be sparse. Recall that

$$\mathcal{L}_{\text{sparse}}(\mathbf{c}) = -\log\|\mathbf{c}\| \quad . \tag{1}$$

' where $\|c\|$ is the L2 norm which is convex. Recall that $c$, the confidence vector, defines a polyhedron, since it is the set of points that are non-negative, and whose element sum up to one. The maximum of a convex function over a polyhedra must occur at the vertices, which correspond to an assignment of 1 to a single $c_i$ and 0s to every other value of $\mathbf{c}$. Next we consider the minimum of $\|c\|$ given that it's elements sum up to one. This is equivalent to finding the stationary points of the Legragian

$$\sum_i c_i^2 + \lambda(\sum_i c_i - 1) \tag{2}$$

By taking the gradient of the above expression, we find that the stationary value corresponds to each $c_i$ being equal. Since the function is convex, this corresponds to the minimum of $\|c\|$. Thus $\mathcal{L}_{\text{sparse}}(\mathbf{c})$ is maximized when each individual confidence is equal.

## A.6 PERFORMANCE USING MORE CLUSTERS

We measure the performance DAF-Net in the presence of a large number of clusters and slots. We consider the *Normal* distribution setting, where input observations are generated by a total of 30 difference clusters. We train DAF-Net with 50 observations, and measure performance at inferring cluster centers with either 50 or 100 observations. We report performance in Table 7 and find that DAF-Net approach obtains good performance in this setting, out-performing both online and offline baselines.

| Type | Model | Online | Observations | | | |
|---|---|---|---|---|---|---|
| | | | 10 | 30 | 50 | 100 |
| Normal | DAF-Net | + | 0.235 (0.001) | 0.162 (0.001) | 0.146 (0.001) | 0.128 (0.001 |
| | Set Transformer | + | 0.390 (0.002) | 0.388 (0.002) | 0.388 (0.002) | 0.389 (0.001) |
| | LSTM | + | 0.288 (0.001) | 0.260 (0.001) | 0.269 (0.001) | 0.288 (0.001) |
| | VQ | + | 0.246 (0.001) | 0.172 (0.001) | 0.147 (0.001) | 0.122 (0.001) |
| | Set Transformer | + | 0.295 (0.003) | 0.261 (0.001) | 0.253 (0.001) | 0.247 (0.001) |
| | K-means++ | - | 0.183 (0.002) | 0.107 (0.001) | 0.086 (0.001) | 0.066 (0.001) |
| | GMM | - | 0.189 (0.002) | 0.118 (0.001) | 0.087 (0.001) | 0.067 (0.001) |
| Mixed | DAF-Net | + | 0.255 (0.002) | 0.184 (0.001) | 0.164 (0.001) | 0.147 (0.001) |
| | LSTM | + | 0.306 (0.002) | 0.274 (0.001) | 0.284 (0.001) | 0.290 (0.001) |
| | Set Transformer | + | 0.415 (0.002) | 0.405 (0.001) | 0.407 (0.001) | 0.408 (0.001) |
| | VQ | + | 0.262 (0.002) | 0.192 (0.001) | 0.169 (0.001) | 0.145 (0.001) |
| | Set Transformer | - | 0.309 (0.002) | 0.274 (0.001) | 0.266 (0.001) | 0.261 (0.001) |
| | K-means++ | - | 0.206 (0.003) | 0.135 (0.001) | 0.105 (0.001) | 0.088 (0.001) |
| | GMM | - | 0.212 (0.003) | 0.136 (0.001) | 0.105 (0.001) | 0.079 (0.001) |
| Enlongated | DAF-Net | + | 0.258 (0.002) | 0.192 (0.001) | 0.173 (0.001) | 0.161 (0.001) |
| | LSTM | + | 0.314 (0.003) | 0.274 (0.002) | 0.288 (0.001) | 0.300 (0.001) |
| | Set Transformer | + | 0.394 (0.003) | 0.391 (0.003) | 0.394 (0.003) | 0.394 (0.003) |
| | VQ | + | 0.265 (0.003) | 0.194 (0.002) | 0.172 (0.001) | 0.149 (0.001) |
| | Set Transformer | - | 0.309 (0.002) | 0.244 (0.002) | 0.240 (0.001) | 0.232 (0.001) |
| | K-means++ | - | 0.213 (0.002) | 0.139 (0.001) | 0.113 (0.001) | 0.092 (0.001) |
| | GMM | - | 0.214 (0.002) | 0.141 (0.001) | 0.112 (0.001) | 0.086 (0.001) |
| Rotation | DAF-Net | + | 0.892 (0.001) | 0.794 (0.001) | 0.749 (0.002) | 0.736 (0.001) |
| | LSTM | + | 0.799 (0.003) | 0.796 (0.002) | 0.795 (0.002) | 0.794 (0.002) |
| | Set Transformer | + | 0.793 (0.003) | 0.794 (0.002) | 0.782 (0.002) | 0.782 (0.002) |
| | VQ | + | 0.956 (0.003) | 1.00 (0.003) | 1.00 (0.003) | 0.984 (0.003) |
| | Set Transformer | - | 0.815 (0.003) | 0.784 (0.002) | 0.779 (0.002 | 0.772 (0.002) |
| | K-means++ | - | 0.827 (0.004) | 0.834 (0.003) | 0.823 (0.002) | 0.802 (0.001) |
| | GMM | - | 0.842 (0.004) | 0.875 (0.001) | 0.867 (0.003) | 0.848 (0.002) |
| Noise | DAF-Net | + | 0.375 (0.001) | 0.343 (0.001) | 0.338 (0.001) | 0.334 (0.001) |
| | LSTM | + | 0.419 (0.001) | 0.406 (0.001) | 0.405 (0.001) | 0.407 (0.001) |
| | Set Transformer | + | 0.434 (0.001) | 0.424 (0.001) | 0.425 (0.001) | 0.424 (0.001) |
| | VQ | + | 1.479 (0.002) | 0.948 (0.002) | 0.826 (0.001) | 0.720 (0.001) |
| | Set Transformer | - | 0.436 (0.001) | 0.407 (0.002) | 0.398 (0.001) | 0.394 (0.001) |
| | K-means++ | - | 1.836 (0.002) | 1.271 (0.002) | 1.091 (0.002) | 0.913 (0.002) |
| | GMM | - | 1.731 (0.002) | 1.215 (0.002) | 1.056 (0.002) | 0.856 (0.002) |

Table 6: Comparison of performance under different settings after training on different distribution for a thousand iterations. We use a total of 3 components, and train models with 30 observations. We report standard error in parentheses.

| Model | Online | Observations | | | |
|---|---|---|---|---|---|
| | | 50 | 65 | 80 | 100 |
| DAF-Net | + | **0.158 (0.001)** | **0.154 (0.001)** | **0.151 (0.001)** | **0.147 (0.001)** |
| VQ | + | 0.162 (0.001) | 0.157 (0.001) | 0.153 (0.001) | 0.148 (0.001) |
| K-means++ | - | **0.155 (0.001)** | 0.151 (0.001) | **0.148 (0.001)** | **0.146 (0.001)** |
| GMM | - | 0.156 (0.001) | 0.151 (0.001) | 0.149 (0.001) | 0.147 (0.001) |

Table 7: Comparison of performance on Normal distribution. We use 30 components, and train models with 50 observations. We report standard error in parentheses. Each cluster observation and center is drawn between -1 and 1, with reported error as the L2 distance between predicted and ground truth means.

## A.7 DISTRIBUTIONS DETAILS

We provide detailed quantitative values for each distribution below. Gaussian centers are drawn uniformly from -1 to 1.

| Model | Learned | Observations | | | |
|---|---|---|---|---|---|
| | | 10 | 30 | 50 | 100 |
| DAF-Net | + | **13.207 (0.003)** | **13.214 (0.004)** | **18.318 (0.006)** | **14.743 (0.003)** |
| LSTM | + | 15.076 (0.008) | 15.132 (0.006) | 19.134 (0.004) | 16.071 (0.006) |
| K-means (Pixel) | - | 18.378 (0.005) | 18.817 (0.005) | 20.269 (0.005) | 18.407 (0.006) |
| K-means (Learned) | + | 31.127 (0.003) | 31.127 (0.003) | 31.127 (0.003) | 31.127 (0.003) |

Table 8: Comparison of performance on regressing the ground truth un-rotated version of an MNIST digit where digits move over time. For DAF-Net, LSTM and K-means (Learned) we use a convolutional encoder/decoder trained on the data; for K-means (Pixel) there is no encoding. We use a total of 3 components and train models with 30 observations. We report MSE error with respect to ground truth unrotated images.

1. *Normal*: Each 2D Gaussian has standard deviation 0.2.
2. *Mixed*: Each distribution is a 2D Gaussian, with fixed identical variance across each individual dimension, but with the standard deviation of each distribution drawn from a uniform distribution from (0.04, 0.4).
3. *Elongated*: Each distribution is a 2D Gaussian, where the standard deviation along each dimension is drawn from a uniform distribution from (0.04, 0.4), but fixed across distributions.
4. *Angular*: Each distribution is a 2D Gaussian with identical standard deviation across dimension and distribution, but points above $\pi$ are wrapped around to $-\pi$ and points below $-\pi$ wrapped to $\pi$. Gaussian means are selected between $(-\pi, -2\pi/3)$ and between $(2\pi/3, \pi)$. The standard deviation of distributions is $0.3 * \pi$.
5. *Noise*: Each distribution has 2 dimensions parameterized by Gaussian distributions with standard deviation 0.5, but with the values of the remaining 30 dimensions drawn from a uniform distribution from $(-1, 1)$.

## A.8  DYNAMIC IMAGES

We further compare DAF-Net with other baselines in the setting where the rendered images move over time. We follow the same setup described in image-based domain, but now consider the MNIST setup with each digit centered at a random position in the image (with parts of a digit that are outside of the image wrapped around to the other side of the image). At each timestep, the center of each moves with a constant velocity, with the goal to predict the un-rotated image at the current center of digit. We report results in Table 8 and find that our approach performs well in this setting also.

## A.9  MODEL/BASELINE ARCHITECTURES

| Dense → h |
|---|
| Dense → h |
| LSTM(h) |
| Dense → h |
| Dense → output |

(a) The model architecture of the LSTM baseline. The hidden dimension $h$ used is 96 for synthetic Gaussian distributions and 128 for Image datasets. For image experiments, the first 2 and last 2 fully connected layers are replaced with image encoders and decoders.

| Dense → h |
|---|
| Dense → h |
| Set Transformer Encoder |
| Set Transformer Decoder |

(b) The model architecture of the Set Transformer baseline. The hidden dimension $h$ is 48 for the synthetic Gaussian distributions. We use the encoder and decoder defined in (Lee et al., 2018) with 4 heads and hidden dimension $h$.

| Dense → h |
|---|
| Dense → h |
| DAF-Net Memory |
| Dense → h |
| Dense → output |

(c) The model architecture of DAF-Net. The hidden dimension $h$ is 64 is for synthetic Gaussian distributions and 128 for the image experiments. We detail in component of the memory of DAF-Net memory below. For image experiments, the first 2 and last 2 fully connected layers are replaced with image encoders and decoders.

Figure A5: Architecture of different models.

| 5x5 Conv2d, 32, stride 2, padding 2 |
| --- |
| 3x3 Conv2d, 64, stride 2, padding 1 |
| 3x3 Conv2d, 64, stride 2, padding 1 |
| 3x3 Conv2d, 64, stride 2, padding 1 |
| 3x3 Conv2d, 128, stride 2, padding 1 |
| Flatten |
| Dense → h |

(a) The model architecture of the convolutional encoder for image experiments.

| Dense → 4096 |
| --- |
| Reshape $(256, 4, 4)$ |
| 4x4 Conv2dTranspose, 128, stride 2, padding 1 |
| 4x4 Conv2dTranspose, 64, stride 2, padding 1 |
| 4x4 Conv2dTranspose, 64, stride 2, padding 1 |
| 4x4 Conv2dTranspose, 64, stride 2, padding 1 |
| 3x3 Conv2d, 3, stride 1, padding 1 |

(b) The model architecture of the convolutional decoder for image experiments.

Figure A6: Architectures of encoder and decoder models on image experiments.

We provide overall architecture details for LSTM in Figure A5a, for the set-transformer in Figure A5b and DAF-Net in Figure A5c. For image experiments, we provide the architecture of the encoder in Figure A6a and decoder in Figure A6b. Both LSTM, DAF-Net, and autoencoding baselines use the same image encoder and decoder.

In DAF-Net memory, the function update$(s_k, n_k, e)$ is implemented by applying a 2 layer MLP with hidden units $h$ which concatenates the vectors $s_k, n_k, e$ as input and outputs a new state $u_k$ of dimension $h$. The value $n_k$ is encoded using the function $\frac{1}{1+n_k}$, to normalize the range of input to be between 0 and 1. The function attend$(s_k, n_k, e)$ is implemented in an analogous way to update, using a 2 layer MLP that outputs a single real value for each hypothesis slot.

For the function relevance$(s_k, n_k, e)$, we apply $NN_1$ per hypothesis slot, which is implemented as a 2 layer MLP with hidden units $h$ that outputs a intermediate state of dimension $h$. $(s_k, n_k, e)$ are fed into $NN_1$ in an analogous manner to update. $NN_2$ is applied to average of the intermediate representations of each hypothesis slot and is also implemented as a 2 layer MLP with hidden unit size $h$, followed by a sigmoid activation. We use the ReLU activation for all MLPs.

## A.10 BASELINE DETAILS

All baseline models are trained using prediction slots equal to the ground truth of components. To modify the set transformer to act in an online manner, we follow the approach in (Santoro et al., 2018) and we apply the Set Transformer sequentially on the concatenation of an input observation with hypothesis slots. Hypothesis slots are updated based off new values of the slots after applying self-attention (Set Transformer Encoder). We use the Chamfer loss to train baseline models, with confidence set to 1.

## A.11 ABLATION

We investigate ablations of our model in Table 9. We ablate the components of sparsity loss, learned memory update, suppression of attention weights and relevance weights. We find that each component of our model contributes to improved performance.

| Sparsity | Learned Memory | Supression | Relevance | Observations | | | |
|---|---|---|---|---|---|---|---|
| | | | | 10 | 30 | 50 | 100 |
| − | − | − | − | 0.382 (0.003) | 0.452 (0.003) | 0.474 (0.003) | 0.487 (0.003) |
| + | − | − | − | 0.384 (0.001) | 0.412 (0.001) | 0.423 (0.001) | 0.430 (0.003) |
| + | + | − | − | 0.335 (0.002) | 0.357 (0.002) | 0.366 (0.003) | 0.387 (0.001) |
| + | + | + | − | 0.279 (0.001) | 0.274 (0.001) | 0.278 (0.001) | 0.282 (0.001) |
| + | + | + | + | 0.238 (0.001) | 0.157 (0.001) | 0.137 (0.001) | 0.131 (0.001) |

Table 9: We ablate each components of DAF-Net on the *Normal* distribution . When learned memory is ablated, DAF-Net updates states based on observed values (appropriate in the Normal Distribution dataset). We report $L_{cluster}$ of predictions and report standard error in parentheses. We find that each proposed component of our model is important for improved performance.

