# OpenReview forum: "Learning Online Data Association"
_ICLR.cc/2021/Conference — Reject_

### Official Review · AnonReviewer2 · 2020-10-27
**Novelty is unclear and experiments are not fully convincing**

**Rating:** 4
**Confidence:** 4

**Review:**

Paper Summary:
This paper proposes a deep neural network for online data association. Specifically, the proposed network includes an encoder, attention module, transitional model, and an output decoder, which can associate each observation with the existing hypothesis. The proposed method is evaluated on a few toy datasets and two small-scale image datasets.

Paper Strengths:
1. The technical explanation is very clear to me and seems sound
2. The experiments are extensive, where the proposed method has been evaluated on three toy datasets and two image domain datasets

Paper Weaknesses:
1. Overall, I feel the presentation of the paper’s contributions is not very clear. What is the core novelty of this paper, conceptually or technically? For example, what are the differences of the proposed method versus deep Kalman filter, deep state-space model (SSM) [A1]? There should be a detailed discussion about the differences with prior work in related work, which however I could not find. In the last paragraph of the related work, the paper ends with “We view our work as an instance of this approach”, but provides no comparison with other instances of this type of approach which are the closest to the proposed method.
2. Although I appreciate that this paper has provided extensive experiments, the data used in the experiments seem to be generated randomly (e.g., one thousand normal distribution), which is hard to be reproduced and compared by the follow-up work. Although I am not familiar with the clustering literature, I believe there are public benchmark datasets where I would suggest to evaluate the proposed method, in order to increase the reproducibility and fairness of the comparison. Also, the image-based datasets (MNIST, Airplane) used for evaluation are too simple. It would be nice to see the proposed method is evaluated on real-world datasets for online data association such as MOT Challenges, which makes the results more convincing.
3. In Table 1, the proposed method seems to perform worse than classical methods K-means++ and GMM, it would be nice to see some analysis to explain why this is the case. Also, it would be nice to add a column for the mean of the results over 5 subsets in Table 2 for an overall understanding of the performance
4. In the abstract, the paper criticizes prior work is computationally expensive, which is not really convincing. Classical methods like Kalman filter can run very fast without GPU, while the proposed method is a neural network-based approach, which I guess is slower than classical Kalman filter, though there is no inference speed analysis in the main paper.
5. In the last paragraph of the introduction, this paper mentions that it is possible to solve the target problem by training a standard RNN, but without discussing what are the disadvantages of this RNN-based approach. As a result, there is no justification and motivation for the proposed approach
6. In the related work, this paper mentions that there is little work in the area of learning for data association, which is not true. For example, in the domain of the visual multi-object tracking domain, there are approaches designed for deep or end-to-end data association such as [A2-A5]. There should be a discussion with such approaches.

Justification:
My decision is made mainly because I feel there lacks a detailed discussion about the differences/novelty over prior work and also I am not fully convinced by the experimental results especially the data that is used for evaluation

References

[A1] Hafner et al. Learning latent dynamics for planning from pixels. ICML 2019

[A2] Xu et al. How to Train Your Deep Multi-Object Tracker. CVPR 2020

[A3] Ma et al. Deep Association: End-to-End Graph-Based Learning for Multiple Object Tracking with Conv-Graph Neural Network. ICMR 2019

[A4] Sun et al. Deep Affinity Network for Multiple Object Tracking. TPAMI 2019.

[A5] D. Frossard and R. Urtasun. End-to-End Learning of Multi-Sensor 3D Tracking by Detection. ICRA 2018.

[A6] Dendorfer et al. MOTChallenge: A Benchmark for Single-camera Multiple Target Tracking. IJCV 2020

---

> ### Author Response · Authors · 2020-11-21
> **Response (1/2)**
>
> Thank you for the detailed review.  We hope we can clarify some points about the paper.
>
> Q1) Overall, I feel the presentation of the paper’s contributions is not very clear. What is the core novelty of this paper, conceptually or technically? For example, what are the differences of the proposed method versus deep Kalman filter, deep state-space model (SSM) [A1]? There should be a detailed discussion about the differences with prior work in related work, which however I could not find. In the last paragraph of the related work, the paper ends with “We view our work as an instance of this approach”, but provides no comparison with other instances of this type of approach which are the closest to the proposed method.
>
>
> The core novelty of our paper is the design and implementation of a method for doing data-association filtering for data sequences that do not have temporally dense object detections, without the prior specification of domain models, by training the system on data from related problems, allowing it to operate in problems where the input and output spaces are not specified in advance.
>
> A fundamental difference between our problem and the one solved by a Kalman filter is that a KF cannot handle data-association problems:  it assumes fixed-dimensional state and observation spaces.  By contrast, a data-association filter maintains hypotheses about several “factors” or “objects” in the latent state;  it can handle different numbers of objects and observations and is permutation invariant over the stored hypotheses.  The deep state-space model proposes a model-based RL method for continuous control tasks.  Although there is a state estimator involved in the controller, there is no data association problem, just as there is no data association problem in a classic Kalman filter.
>
> The general approach, of which we view our method to be an instance, is the conceptual framework of taking a classical model-based algorithm, forming a differentiable version of it with flexible neural-networks in place of the classical models, and training the overall structure end-to-end.  We do not know of anyone having applied this overall strategy to the problem of multi-object data-association filtering, and so there is no direct empirical comparison to be made.
>
>
> Q2)  It would be nice to see the proposed method is evaluated on real-world datasets for online data association such as MOT Challenges, which makes the results more convincing.
>
> The suggestion to run our method on and to discuss in more detail the work related to the MOT challenge-type problems is a reasonable one.  We will make clearer in the paper the ways in which the problem class we are addressing differs from that one.   In particular,
> We do not assume relatively dense tracking-type observation sequences.  Any individual “object” may go unobserved for a long time, and then appear again sporadically throughout the observation sequence.  This matches the type of experience a robot might have in observing objects as it moves through a house, which is distributionally very different from the MOT image-based tracking problems.
> We do not assume anything specific about the input or output spaces (whereas MOT solutions generally assume image fragment inputs and image-space tracks as output);  the DAFnet architecture effectively learns how to aggregate observations from one space into a final hypothesis in another space.  This is illustrated in our image-based experiments, in which observations are made from substantially different viewpoints but aggregated into a canonical output form.
>
>
> We designed our image-based experiments to test performance in this new type of problem, for which we are not aware of any existing real-world data sets.  It is a substantial, but very desirable, piece of future work to construct such data sets, ideally from real robot experience.

---

> > ### Author Response · Authors · 2020-11-21
> > **Response (2/2)**
> >
> > Q3) In Table 1, the proposed method seems to perform worse than classical methods K-means++ and GMM, it would be nice to see some analysis to explain why this is the case. Also, it would be nice to add a column for the mean of the results over 5 subsets in Table 2 for an overall understanding of the performance
> >
> > Table 1 reports our results in the simplest possible relevant domain, of online clustering of Gaussian data in a Euclidean space.   This is a problem class  for which *K-means and GMM were specifically hand designed*.  Furthermore, those methods are allowed to run in batch-mode, which gives them an additional advantage.  They should be viewed as a probably-unattainable upper bound on the performance of online methods.
> >
> > One reason for not averaging over the results in table 2 is that the test domains have very different attributes:  in particular, the angular data is in a space where the Euclidean metric is inappropriate and the “noise” data are very high-dimensional.  These domains were specifically constructed by us to violate the built-in assumptions of classical clustering methods, and so it is appropriate to study each case separately to see how aspects of the task distribution affect the accuracy of each method.
> >
> > Q4) In the abstract, the paper criticizes prior work is computationally expensive, which is not really convincing. Classical methods like Kalman filter can run very fast without GPU, while the proposed method is a neural network-based approach, which I guess is slower than classical Kalman filter, though there is no inference speed analysis in the main paper.
> >
> > It is true that Kalman filters are very efficient to run, but our problem class is not solvable by a Kalman filter.  General data-association filters are generally much more computationally complex because of a combinatorial explosion of possible associations of observations to hypotheses.  By learning to do the data association, we hope to be able to perform accurate filtering with a relatively smaller number of hypotheses, and thus less computation.
> >
> > Q5) In the last paragraph of the introduction, this paper mentions that it is possible to solve the target problem by training a standard RNN, but without discussing what are the disadvantages of this RNN-based approach. As a result, there is no justification and motivation for the proposed approach
> >
> >  We do, in fact, compare our method to a recurrent neural network throughout the paper:  we use the LSTM form of RNN because it has generally shown to be the most effective form of RNN for retaining information over long time-horizons with little degradation.
> >
> > Q6) In the related work, this paper mentions that there is little work in the area of learning for data association, which is not true. For example, in the domain of the visual multi-object tracking domain, there are approaches designed for deep or end-to-end data association such as [A2-A5]. There should be a discussion with such approaches.
> >
> > It is true that we “mis-spoke” when we said that there was very little work in learning for data association.  There is indeed a lot of work on data association *for the specific case of visual MOT-type domains*, which we discuss in the following paragraph of the paper;  we will add the citations that you kindly provided.   There is, however, very little work in the more general case, which differs in assumptions as we outline above.

---

### Official Review · AnonReviewer3 · 2020-10-27
**potentially important direction**

**Rating:** 6
**Confidence:** 2

**Review:**

The paper proposes DAF-Net, an attention-based architecture for online filtering, which is used for a flexible set of tasks: online clustering, tracking, and image association.

**Strength**: I like the idea of using attention for updating temporal beliefs, which is natural and potentially a backbone for a wide range of temporally extended tasks. The architecture looks reasonable and handles observation encoding, dynamics updating, and online association altogether, and the presented numbers look good across three experiments.

**Weakness**: See **Clarity** and **Questions**.

**Clarity**: I find the paper not fully accessible at the first sight.
- Section 3 "Problem Formulation" is quite abstract and unintuitive, and some motivation before description might be good. I'm not sure why the loss function is part of problem formulation rather than the method.
- Section 4 "DAF-Nets" presents some parts I do not understand, e.g. the format of $n$ vector, and the sentence "The $a$ vectors are integrated to obtain $n$, which is normalized to obtain the final output confidence values $c$". The use of $s_k$ and $s_{tk}$ are inconsistent.
- Figure 1 looks complicated and the meaning of colors of edges are not annotated.
- Section 5 "Empirical Results" is informative, but in general hard to access. To start with, the first two tasks are described in bulks of text, and I believe any visualization would be valuable for intuition (esp. task 2). Tables lack bolded parts for readability, and I'd appreciate if they can also label what models are trained and what models are unsupervised (though it's in text).

**Questions**:
- How to handle multi-object tracking when the observation encoding is just one piece, but not object-factorized? 3 objects are still okay but I guess more objects would be intractable?
- Task 1 and 3 are not really problems with a temporal line. So why is this online approach *supposed* to be better than the batch approach? I know the empirical result is better but I still want to understand it conceptually.
- I'm not sure if baselines are strong or designed for these tasks, especially task 3 seems a bit ad hoc. It might be more interesting to render MNIST and airplane into tracking tasks if the aim is to show ability to handle high-dim inputs.

**Originality**: I'm not familiar with related work but the method looks original, possibly as the abstract problem formulation is novel.

---

> ### Author Response · Authors · 2020-11-21
>
> We are sorry that aspects of the paper were not clear.
>
> Q1) Section 3 "Problem Formulation" is quite abstract and unintuitive, and some motivation before description might be good. I'm not sure why the loss function is part of problem formulation rather than the method.
>
> The loss function is part of the problem formulation because we are defining a new class of problem, and specifying an evaluation metric that *any* solution to this problem should be measured by.
>
> Q2) Section 4 "DAF-Nets" presents some parts I do not understand, e.g. the format of n vector, and the sentence "The a vectors are integrated to obtain n, which is normalized to obtain the final output confidence values c".
>
> We were not clear enough in our description of n and c, and we have clarified in the paper revision.  At any time step t, a_t represents the vector of attention values that each slot has given to the observation at time t.   These a_t values are summed (“integrated”)  over time, so that n_t represents the cumulative sum of attention that each slot has received up to time t.  The n_t vector is put through a softmax transformation to obtain the final confidence values c.
>
> Q3) The use of sk and stk are inconsistent.
>
> It is true that st and stk are used somewhat inconsistently.   Our intent was that:
> a subscript of k always selects a slot
> when there is no ambiguity about time step, s stands for all the slot values;  so s_k is a particular current slot value
> when we are describing the dynamics, we add an explicit subscript t for the time-step of the filtering process (we could add it everywhere, but thought it would make things even more confusing).  In that case:
> s_t is the current state of all the slots (so s_tk is the current state of a particular slot)
> s’_t is the state of all the slots when they have had the observation factored in
> NN_3(s’_t) is the predicted state of all the slots, based on the slot values at time t and the observation at time t
> this will be the new set of slot values, so s_{t+1} = NN_3(s’_t)
> Writing this out in detail revealed to us a small bug in the figure (one arrow was mis-oriented, and so we have corrected that in the new version of the paper).
>
> Q4) Figure 1 looks complicated and the meaning of colors of edges are not annotated.
>
> The colors of the arrows in the figure are just intended to help “trace” the wires---so all of the red lines carry the same signal (same for green and blue).  We can clarify in the text.  We agree that it is somewhat complex but do not know how to make it simpler and correct at the same time.
>
> Q5) Section 5 "Empirical Results" is informative, but in general hard to access. To start with, the first two tasks are described in bulks of text, and I believe any visualization would be valuable for intuition (esp. task 2). Tables lack bolded parts for readability, and I'd appreciate if they can also label what models are trained and what models are unsupervised (though it's in text).
>
> We have added illustrations of both clustering and dynamics in figure 2XXX. To improve clarity we have  also bolded tables with the best values and have also labeled models as unsupervised or supervised in tables.
>
> Q6) How to handle multi-object tracking when the observation encoding is just one piece, but not object-factorized? 3 objects are still okay but I guess more objects would be intractable?
>
> If the observations are not in any way segmented, then this approach doesn’t apply directly.  It is in principle possible to compose a DAFnet with a learned segmentation algorithm and train the whole system end-to-end, though that would require substantially more training data and time.
>
> Q7) Task 1 and 3 are not really problems with a temporal line. So why is this online approach supposed to be better than the batch approach? I know the empirical result is better but I still want to understand it conceptually.
>
> The online approach to estimation (in which the system produces an estimate of the current latent world state based on observations up until the current time), but not on future observations, and in which the computation needs to be efficiently executable online, is not expected to make more accurate predictions than the batch approach.  We study it because many kinds of real-world systems require online predictions computed efficiently based only on current data.  DAFNet performs better than set transformer because it has a strong inductive bias that is appropriate for this problem, whereas set transformer is more general-purpose, and so DAFNet can learn more effectively with a small amount of data.
>
> Q8) I'm not sure if baselines are strong or designed for these tasks, especially task 3 seems a bit ad hoc. It might be more interesting to render MNIST and airplane into tracking tasks if the aim is to show ability to handle high-dim inputs.
>
> It is true that a dynamic version of the image domain would be interesting.   We are implementing it now and expect to have results soon.

---

> > ### Author Response · Authors · 2020-11-25
> > **Dynamic Results**
> >
> > We have added results with a version of MNIST where the digits dynamically move and rotate or time in appendix A.8.

---

### Official Review · AnonReviewer4 · 2020-10-28
**Good method but the experiment settings are not practical**

**Rating:** 6
**Confidence:** 3

**Review:**

This work aims to solve the data association problem in a general setting, which is suitable for various scenarios such as online clustering and online tracking. The motivation of having this setting is that temporal dense observations are not always available.

The high level idea of this work is to learn a neural network which updates the hypothesis ($y$) and outputs the association ($c$). I like the design that are well motivated by the problem itself and the ability of neural networks: 1) recursive filter for online data; 2) order invariant memory structure; 3) representation learning.

The experiments are conducted on many domains and the proposed approach is compared with many methods from different domains. Ablation study is also conducted to demonstrate the effectiveness of the design decisions. It seems thorough. However, I think the experiments are more like toy experiments and it would be great if it could demonstrate its superiority on a more practical dataset such as a multiple object tracking (MOT) benchmark. In my opinion, data association is the key for MOT. But definitely other problems with a practical setting can also convince me. Another concern about the experiment is that the loss is used as the metric, which seems weired to me. The clustering metrics like F-measure or Rand index and/or precision and recall might be better.


Other suggestions and clarifications:

1. I am very interested in the sparsity loss. Is there any references or justifications which shows that it will leads to a sparse result.

2. Is it possible to use a classification (cross entropy) loss for $c$? Will the order invariant property cause any issue here?

3. An additional question about $c$: how is it normalized?

4. Why "the basic clustering methods have no ability to handle dynamic systems"? I assume they don't have the dynamic part, but as long as they can observe the object `"feature", they are applicable. Though I guess the performance won't be good.

5. In the image-based domains, is it generally better than the classification methods?

Missing reference:
Guillem Braso and Laura Leal-Taixe. Learning a Neural Solver for Multiple Object Tracking. In CVPR 2020. This work aims to solve the data association problem for Multiple Object Tracking based on Message Passing Networks.

=== Post rebuttal ===
1. As reviewer 2, I am also not convinced by the reason why it could not be evaluated on a MOT dataset. The explanation suggests that the proposed method is more general and it should be able to apply it to broader domains (and thus MOT could be one of them).
2. The evaluation metric still looks suspicious to me. I can imagine we can usually get better numbers if we can directly minimize the metrics we want to evaluate. However, in my opinion, data assocision usually need to do hard assginments if we want to use it in practice (unless it is a intermediate task and in that case we can do soft assignments), so I do think it is better to set a hard threshold or draw a curve of F1 v.s. threhold. In the meantime,  I think it is unfair for some other methods, like k-means++ becaue it is minimizing a different objective.

I still think the method has some merits so I am at borderline, though I will not defend it if it is rejected.

---

> ### Author Response · Authors · 2020-11-21
> **Response**
>
> Thank you for your comments.
>
> Q1) However, I think the experiments are more like toy experiments and it would be great if it could demonstrate its superiority on a more practical dataset such as a multiple object tracking (MOT) benchmark. In my opinion, data association is the key for MOT. But definitely other problems with a practical setting can also convince me.
>
> We definitely understand and appreciate the desire for experiments on a real-world data-set.   However, it is important to observe that we are addressing a problem here that is fundamentally different in its underlying assumptions from the MOT-style challenge problems in the following ways:
> We do not assume relatively dense tracking-type observation sequences.  Any individual “object” may go unobserved for a long time, and then appear again sporadically throughout the observation sequence.  This matches the type of experience a robot might have in observing objects as it moves through a house, which is distributionally very different from the MOT image-based tracking problems.
> We do not assume anything specific about the input or output spaces (whereas MOT solutions generally assume image fragment inputs and image-space tracks as output);  the DAFnet architecture effectively learns how to aggregate observations from one space into a final hypothesis in another space.  This is illustrated in our image-based experiments, in which observations are made from substantially different viewpoints but aggregated into a canonical output form.
>
> We designed our image-based experiments to test performance in this new type of problem, for which we are not aware of any existing real-world data sets.  It is a substantial, but very desirable, piece of future work to construct such data sets, ideally from real robot experience.
>
> Q2) Another concern about the experiment is that the loss is used as the metric, which seems weired to me. The clustering metrics like F-measure or Rand index and/or precision and recall might be better.
>
> The reported errors are Cartesian distances from the estimated mean produced by the network and the ground-truth mean.   We use this objective, both as a loss for training and as our evaluation metric, because the overall goal of the problem is to recover properties of the underlying objects whose state we are estimating, rather than to recover the observation distribution or to determine an assignment of observations to objects (which are the objectives in clustering, measured typically via KL-distance and RAND score, respectively).  It’s not completely clear how to apply precision/recall/f-measure to our problem:  we could potentially say that if one of our hypotheses is within some fixed error to one of the ground-truth objects then we have accurately “retrieved” it.
>
> Q3) I am very interested in the sparsity loss. Is there any references or justifications which shows that it will leads to a sparse result.
>
> We are not aware of previous uses of this form of sparsity loss.  We have added a theoretical characterization of the sparsity loss in section A.5.  We show that it is minimized when the confidence c is evenly distributed across all slotsand maximized when a single slot is assigned full confidence.  We take the log to amplify difference between the maximum and minimum sparsity loss.
>
> Q4) Is it possible to use a classification (cross entropy) loss for c? Will the order invariant property cause any issue here? An additional question about c: how is it normalized?
>
> The output c is intended to reflect confidence in the hypothesis contained in the slot.  It is part of the mechanism of the DAFNet, rather than an output for which any type of supervision could be expected to be available.   It might be possible to construct an auxiliary loss for c, in which we want it to be high in cases where the slot’s hypothesis matches some true object and low otherwise;  but training this way would conflict with the desire for sparsity.
>
> We are sorry not to have been clear:  the c values are computed using a softmax transformation on the n values, which represent the sum of “responsibilities” so far, for each slot.
>
> Q5) Why "the basic clustering methods have no ability to handle dynamic systems"? I assume they don't have the dynamic part, but as long as they can observe the object `"feature", they are applicable. Though I guess the performance won't be good.
>
> Yes, the clustering methods meet the input/output specification of the dynamic problems, but perform very very badly because they do not have the flexibility to allow the means to change over time.
>
> Thank you for pointing out the Braso and Leal-Taixe reference.  It is an interesting paper, but it is fundamentally focused on the MOT-style problems, which are substantially different from ours in their assumptions, as described above.  We have included it in our section on visual data-association methods.

---

### Official Review · AnonReviewer1 · 2020-11-02

**Rating:** 7
**Confidence:** 4

**Review:**

The paper proposes a method to train a deep network to perform dynamic data association and static online clustering. The proposed architecture relies on a recurrent structure while also adding knowledge about the data association problem into the architecture design.

The majority of the paper is well written with an easy to understand and follow motivation and a concise yet complete review of related work. The problem formulation and description of the network are well written and provide a good intuition for the choices made. While some of the choices made in the architecture design are motivated this is not done for all aspects, and it would have been nice to see this as it could provide greater insight into what works and what does not.

The experimental section of the paper is probably the weakest part. This is partly due to the writing getting less clear in certain aspects and some choices made not being clear. Despite this, the methods used in the comparison seem adequate to demonstrate the ability of the proposed method.

The paper states that in the Gaussian domains, the proposed method's performance is comparable to that of the non-online methods such as k-means. Looking at the numbers, there is a difference of 0.5 at times (Table 1). Given the scales of the values, this would appear to be a substantial difference. As these results effectively evaluate clustering performance, the chosen metric seems a bit odd. Usually, one would expect normalized mutual information or similar clustering metrics to be used. Overall it just is not clear what the magnitudes reported represent.

Another aspect that always remained somewhat mysterious was the way the number of observations was used. Is it that the different algorithms were given N observations to process at once or N observations in sequence. In either case, how is the number reported obtain, is it the correctness of every single point in the sequence? While the overall aspect that the different experiments attempt to capture was understandable, the finer details never become entirely clear, which is a shame given the clarity of the rest of the paper.

The paper concludes that the proposed method scales well to an increased number of hypothesis slots and a large number of underlying clusters. I do not see where this conclusion is supported by the experiments if anything the opposite is shown. Table 3 shows what appears to be a relation between the number of slots and true clusters for performance. Though in any case, the performance degrades with an increase of true clusters present. This leads to another question; the experiments seem to focus on a low number of true clusters and a low number of slots. In the robotic scenarios outlined at the beginning of the paper, one would expect there to be hundreds of objects that need to be tracked. This leads to several questions. How does the proposed method handle cases where the number of slots is 100 or more and the number of clusters is similarly high. Furthermore, is there a reason not to use a model with an overly large number of slots, say 1000, to be sure there is sufficient capacity for complex environments?

The image domain based experiments were initially quite confusing, mainly because these datasets are used in a non-traditional setup, i.e. clustering or classification. After reading the explanation carefully twice, I could understand what is being done, though initially, I was quite confused.

=== Post rebuttal ===

The clarification and additional experiments are greatly appreciated and answered some of the points which were not entirely clear to me.

---

> ### Author Response · Authors · 2020-11-21
> **Response (1/2)**
>
> Thank you for your detailed comments. We address major concerns below and have also correspondingly updated the text.
>
> Q1) The paper states that in the Gaussian domains, the proposed method's performance is comparable to that of the non-online methods such as k-means. Looking at the numbers, there is a difference of 0.5 at times (Table 1). Given the scales of the values, this would appear to be a substantial difference. As these results effectively evaluate clustering performance, the chosen metric seems a bit odd. Usually, one would expect normalized mutual information or similar clustering metrics to be used. Overall it just is not clear what the magnitudes reported represent.
>
>
> It was a significant oversight not to clearly report the units in table 1.  The samples in these problems are drawn from a space in which each dimension ranges from -1 to +1, except in the angular case, when it ranges from -pi to +pi.  The reported errors are Cartesian distances from the estimated mean produced by the network and the ground-truth mean.   We use this objective because the overall goal of the problem is to recover properties of the underlying objects whose state we are estimating, rather than to recover the observation distribution (which is the objective in clustering that is measured by mutual information).
>
> Q2) The paper states that in the Gaussian domains, the proposed method's performance is comparable to that of the non-online methods such as k-means. Looking at the numbers, there is a difference of 0.5 at times (Table 1). Given the scales of the values, this would appear to be a substantial difference.
>
> It is true that for the distributions matching its assumptions (normal, elongated, and mixed) that k-means and GMM outperform DAF-Net, because they have access to all the data simultaneously (in batch mode) and they have the correct modeling assumptions baked in.  But in the non-Cartesian (Angular) and high-dimensional (Noise) cases, which violate the standard assumptions, even though they can operate in batch mode they do not perform as well as DAFNet.
>
> Q3) Another aspect that always remained somewhat mysterious was the way the number of observations was used. Is it that the different algorithms were given N observations to process at once or N observations in sequence. In either case, how is the number reported obtain, is it the correctness of every single point in the sequence?
>
> The observations are always fed in sequentially to all algorithms.   We train the learning-based machines on a set of sequences of 30 observations, where each sequence is generated by a different distribution within the problem’s distribution family.  The error reported is the correctness of the final point of the sequence, so results reported in the first column of table 1 is the distance between the predicted means and the ground-truth means after the model has seen the first 10 observations of a new problem.   An important observation is that although the DAFnet models were only ever trained to absorb sequences of 30 observations, they continue perform well for sequences of 50 and 100 observations, with estimation error actually *decreasing*, which implies they have learned to perform a relatively general recursive estimation procedure.
>
> Q4) The paper concludes that the proposed method scales well to an increased number of hypothesis slots and a large number of underlying clusters. I do not see where this conclusion is supported by the experiments if anything the opposite is shown. Table 3 shows what appears to be a relation between the number of slots and true clusters for performance. Though in any case, the performance degrades with an increase of true clusters present.
>
> The experiment in table 3 is meant to test the models’ ability to generalize beyond the types of problems they were trained on.  In all cases, they were trained on problems with 3 ground-truth clusters, and DAF-Net was always trained with 10 slots.   We then test, *with no further training*, on both problems with more true clusters and on DAF-Nets with more slots, and find that the performance does not degrade catastrophically.

---

> > ### Author Response · Authors · 2020-11-21
> > **Response (2/2)**
> >
> > Q5) This leads to another question; the experiments seem to focus on a low number of true clusters and a low number of slots. In the robotic scenarios outlined at the beginning of the paper, one would expect there to be hundreds of objects that need to be tracked. This leads to several questions. How does the proposed method handle cases where the number of slots is 100 or more and the number of clusters is similarly high. Furthermore, is there a reason not to use a model with an overly large number of slots, say 1000, to be sure there is sufficient capacity for complex environments?
> >
> > Your point about extending to larger numbers of objects is a good one.   We have done additional testing with 30 different cluster centers and sequences of 100 observations, and report the results below and in Table 6 of the appendix.   In principle, it could be extended to even larger problem sizes, although eventually additional mechanisms for spatial indexing and-or more strongly enforced sparsity will have to come into play for reasons of both computational efficiency (at training and test time) as well as accuracy.

---

### Decision · Program_Chairs · 2021-01-07
**Final Decision**

**Decision:**

Reject

**Comment:**

This paper was on the borderline. While there was some support for the ideas presented, concerns were raised about the experiments. The exposition would also need to better demonstrate the significance of the contribution.